# Molecular engineering towards efficient white-light-emitting perovskite

Mingming Zhang[1,7], Lili Zhao[2,7], Jiahao Xie [3,7], Qian Zhang[1,7], Xiaoyu Wang[3], Najma Yaqoob[4], Zhengmao Yin[5], Payam Kaghazchi [4], San Zhang[6], Hua Li[2], Chunfeng Zhang [6], Lei Wang [1], Lijun Zhang [3✉], Weigao Xu [2✉] & Jun Xing [1✉]

Low-dimensional hybrid perovskites have demonstrated excellent performance as white-light emitters. The broadband white emission originates from self-trapped excitons (STEs). Since the mechanism of STEs formation in perovskites is still not clear, preparing new low-dimensional white perovskites relies mostly on screening lots of intercalated organic molecules rather than rational design. Here, we report an atom-substituting strategy to trigger STEs formation in layered perovskites. Halogen-substituted phenyl molecules are applied to synthesize perovskite crystals. The halogen-substituents will withdraw electrons from the branched chain ($-R-NH_3^+$) of the phenyl molecule. This will result in positive charge accumulation on $-R-NH_3^+$, and thus stronger Coulomb force of bond ($-R-NH_3^+$)-($PbBr_4^{2-}$), which facilitates excitons self-trapping. Our designed white perovskites exhibit photoluminescence quantum yield of 32%, color-rendering index of near 90 and chromaticity coordinates close to standard white-light. Our joint experiment-theory study provides insights into the STEs formation in perovskites and will benefit tailoring white perovskites with boosting performance.

[1] Key Laboratory of Eco-Chemical Engineering, Ministry of Education, College of Chemistry and Molecular Engineering, Qingdao University of Science & Technology, Qingdao, China. [2] Key Laboratory of Mesoscopic Chemistry, Ministry of Education, School of Chemistry and Chemical Engineering, Nanjing University, Nanjing, China. [3] State Key Laboratory of Integrated Optoelectronics, Key Laboratory of Automobile Materials of MOE, School of Materials Science and Engineering, Jilin University, Changchun, China. [4] Forschungszentrum Jülich GmbH, Institute of Energy and Climate Research, Materials Synthesis and Processing (IEK-1), Wilhelm-Johnen-Straße, Jülich, Germany. [5] School of Materials Science and Technology, Qingdao University of Science & Technology, Qingdao, China. [6] National Laboratory of Solid State Microstructures, School of Physics and Collaborative Innovation Center of Advanced Microstructures, Nanjing University, Nanjing, China. [7] These authors contributed equally: Mingming Zhang, Lili Zhao, Jiahao Xie, Qian Zhang.
✉email: lijun_zhang@jlu.edu.cn; xuwg@nju.edu.cn; xingjun@qust.edu.cn

Artificial lighting consumes about one-fifth of global electricity. The traditional method to produce white light is mixing multiple color emitters, which suffers from several problems, such as unstability in emission color due to different degradation rates of emitters and efficiency loss due to the overlapping absorption. Thus, developing a single-phase material with an efficient broadband white-light (BWL) emission and comparable stable colors as well as good color rendition is ideal for lighting application. In recent years, metal halide perovskites have emerged as a class of material with low-temperature solution processing, tunable band structure and efficient photoluminescence, which offer an intriguing potential application for low-cost light-emitting devices[1–5]. In 2014, Karunadasa et al.[6] observed a BWL emission from a two-dimensional (2D) lead-halide perovskite. Subsequently, a variety of low-dimensional perovskites were demonstrated to be BWL emissive[7–11]. The large Stokes shift and broad photoluminescence (PL) of low-dimensional perovskites originate from self-trapped excitons (STEs), which form through strong coupling of excited excitons with surrounding deformable lattice[12]. Because of unclear mechanism of STEs formation, most of the reported low-dimensional white perovskites have been so far discovered by screening a series of intercalating organic molecules; it is still a big challenge to design new white perovskite materials with boosting BWL emission on atomic level.

STE is a common phenomenon in metal halides, since their ionic and polar crystal structures possess relative strong electron-phonon interaction[13]. Photo-induced excitons in low-dimensional perovskites have large binding energies of several hundreds of meV, in which an electron-hole pair shares common bonds. Owing to the electroneutrality feature of the excitons, STEs are dominated by short-range electron-phonon interactions[14]. Thus, the bonds surrounding the excitons should play an important role in the formation of STEs in perovskites.

Herein, we report an atomic-level tailoring strategy to enhance the Coulomb force of ionic bonds in 2D perovskites. The strengthened bonds trigger excitons self-trapping at room temperature, resulting in an efficient BWL emission from 400 to 800 nm. Our proposed white perovskites offer high PLQY of 32% and high color-rendering index (CRI) of about 90 as well as excellent Commission International de I'Eclairage (CIE) coordinates close to the standard white-light (0.33, 0.33). The white perovskite exhibits good stability under continuous heating at 100 °C and exposing in air for 400 h. This work not only discovers a series of new perovskites emitting broadband white-light with high quantum efficiency and good stability, but unravels the effect of the chemical bonds strength on the STEs formation, and thus offers a useful perspective to design new white perovskite materials.

## Results

### Synthesis and structures of perovskite crystals

Two-dimensional lead-bromide perovskite crystals were synthesized according to the previous reported anti-solvent diffusion method[15,16]. A vial containing $PbBr_2$, $R-NH_3Br$, and DMF/DMSO solution was put in a bigger sealed vial container with anti-solvent. Perovskites crystalized gradually as anti-solvent diffuses into perovskite precursor and the crystals were then taken out after several days (see Methods for details). We took Raman measurements on the perovskites and their corresponding alkylamines precursors to resolve the microstructure. As shown in Supplementary Fig. 1, Raman features of all the perovskites can match well with their corresponding precursors. For the halogen-substituted perovskites, the typical Raman band at about 680 cm$^{-1}$ is attributed to typical halogen-substituent-sensitive vibration[17], which is not observed on

pristine perovskite. Thus, halogen-substituted perovskites were successfully synthesized, and the halogen substituents can stably stay in the perovskite products.

Phenylmethylammonium lead bromide (($PMA)_2PbBr_4$) is a typical violet emissive perovskite without STEs phenomenon at room temperature[18]. To investigate the influence of chemical bonds on the formation of STEs, halogen atoms were introduced as substituents in phenyl to tailor the chemical property of PMA cations. Firstly, single-crystal XRD was conducted to monitor the structural change in halogen-substituted perovskites. Figures 1a–f show the atomic structure of ($PMA)_2PbBr_4$, (2-$FPMA)_2PbBr_4$ (ortho-substitution), (2-$ClPMA)_2PbBr_4$, (2-$BrPMA)_2PbBr_4$, (3-$ClPMA)_2PbBr_4$ (meta-substitution) and (4-$ClPMA)_2PbBr_4$ (para-substitution). As expected, ($PMA)_2PbBr_4$ shows layered perovskite structure with PbBr6 octahedrons connecting with corners, which crystalizes in the polar space group $Cmc2_1$[18]. The interlayer spacing is 1.668 nm. When halogen replaces ortho-hydrogen, the space group of the perovskites keeps orthorhombic system (Fig. 1 and Supplementary Tables 1–3). The crystal lattices of (2-$FPMA)_2PbBr_4$ remain almost the same as those of ($PMA)_2PbBr_4$, which is attributed to the comparable bond length of C-F (1.41 Å) to C-H (1.08 Å). When Cl or Br replace ortho-hydrogen, one of the in-plane-orientation crystal lattices slightly expands due to the longer bond length of C-Cl (1.77 Å) and C-Br (1.91 Å). Meanwhile, the interlayer spacings of (2-$ClPMA)_2PbBr_4$ and (2-$BrPMA)_2PbBr_4$ are compressed to 1.626 and 1.627 nm, respectively. However, in (3-$ClPMA)_2PbBr_4$ and (4-$ClPMA)_2PbBr_4$ perovskites, the interlayers are pushed aside (Supplementary Tables 2 and 3) and the space group of (3-$ClPMA)_2PbBr_4$ transforms into monoclinic system. Powder XRD testing further verified these results. In Fig. 1g, the well-defined diffraction peaks are corresponding to the (00l) reflections series of the layered perovskites. All the diffraction peaks of (2-$FPMA)_2PbBr_4$ and (3-$FPMA)_2PbBr_4$ are very close to those of ($PMA)_2PbBr_4$, and (2-$Cl/BrPMA)_2PbBr_4$ shift to larger degrees in comparison to ($PMA)_2PbBr_4$, indicating a decreased interlayer distance. On the contrary, the diffraction peaks of (4-$FPMA)_2PbBr_4$, (3-$Cl/BrPMA)_2PbBr_4$ and (4-$Cl/BrPMA_2)PbBr_4$ shift to smaller degrees. According to Bragg's diffraction equation ($2d \cdot \sin\theta = n\lambda$), interlayer distances of the perovskites were calculated and summarized in Supplementary Table 4, of which the results are in fair agreement with the single-crystal XRD.

### Photophysical properties of perovskite crystals

Despite the structural similarities between pristine and halogen-substituted ($PMA)_2PbBr_4$, their optical properties are distinct. At room temperature, ($PMA)_2PbBr_4$ displays only a narrow violet emission located at 410 nm (Fig. 2a). However, the PL spectra of (2-F/Cl/$BrPMA)_2PbBr_4$ show broadband emissions from 400 to 800 nm, which are composed of a narrow free-excitons (FEs) peak and a much broad STEs peak (Fig. 2a–c). The CIE chromaticity coordinates of the overall spectra of (2-$FPMA)_2PbBr_4$, (2-$ClPMA)_2PbBr_4$ and (2-$BrPMA)_2PbBr_4$ were determined to be (0.342, 0.361), (0.300, 0.330) and (0.312, 0.338), which give correlated color temperatures (CCT) of 5139, 7215, 6478 K, respectively (Fig. 2d). These locations are very close to the standard white light (0.33, 0.33). They display CRI values of 89, 86 and 88, respectively, which are higher than that of most commercial white-light-emitting diodes (CRI of about 80). The PLQYs of (2-F/Cl/$BrPMA)_2PbBr_4$ crystals were measured to be 5%, 32% and 15%, respectively. The performance is much better than that of most low-dimensional white perovskites reported (Supplementary Table 5). It should be noted that these FEs emissions of (2-F/Cl/$BrPMA)_2PbBr_4$ exhibit an asymmetric band spectra, including one main PL peak and two shoulder peaks. The location of the high-energy shoulder is corresponding to the absorption band

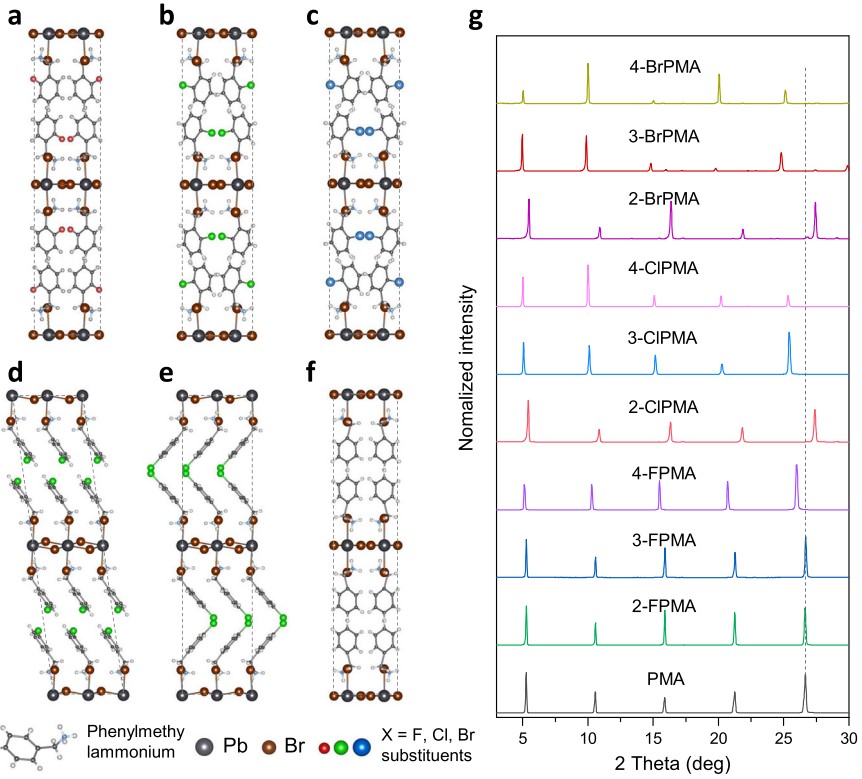

**Fig. 1 Structure of pristine and halogen-substituted perovskites. a–f** Atomic structure of perovskites (2-FPMA)$_2$PbBr$_4$, (2-ClPMA)$_2$PbBr$_4$, (2-BrPMA)$_2$PbBr$_4$, (3-ClPMA)$_2$PbBr$_4$, (4-ClPMA)$_2$PbBr$_4$ and (PMA)$_2$PbBr$_4$, respectively. **g** Powder XRD patterns of perovskites (PMA)$_2$PbBr$_4$, (FPMA)$_2$PbBr$_4$, (ClPMA)$_2$PbBr$_4$ and (BrPMA)$_2$PbBr$_4$.

edge. According to previous studies, this high-energy shoulder peak may arise from the reabsorption of high-energy region of the FEs emission spectrum in thick perovskite[16]. The low-energy shoulder is similar to that reported in low-dimensional perovskites, which might derive from a vibrational replica of the intrinsic band[10,19]. One can see the FEs emission peaks of (F/Cl/BrPMA)$_2$PbBr$_4$ at different locations, which might be determined by multiple factors. As mentioned above, strong reabsorption of high-energy region of the FEs emission would occur in the bulk crystals, which may lead to a dominative PL peak redshifting. The absorption edge of perovskites (F/Cl/BrPMA)$_2$PbBr$_4$ are slightly different with each other, indicating the different bandgap of these perovskites. The Stokes-shift range of the FEs emission of these perovskites might be also different. They codetermine the FEs emission properties.

The STEs origin of the BWL emission was experimentally confirmed by implementing power dependent PL measurement, which shows a linear dependence from 0.024 to 23.5 W cm$^{-2}$ (Fig. 3a). Otherwise, PL from permanent defects usually shows a sublinear dependence on the excited power with a saturation of limited defect sites under high excitation intensity (see Supplementary Note 1 for details)[2,7,8]. We also observed a similar broad spectrum from (2-ClPMA)$_2$PbBr$_4$ powders synthesized by fast reprecipitation method (see Methods for details, Supplementary Fig. 2), which further confirms that the broadband emission does not originate from the defects in crystals. The PL intensity ratios $I_{STEs}/I_{FEs}$ of (2-FPMA)$_2$PbBr$_4$, (2-ClPMA)$_2$PbBr$_4$ and (2-BrPMA)$_2$PbBr$_4$ are approximately 4.1, 2.9, and 2.9, respectively. They are much higher than that of (3-F/Cl/BrPMA)$_2$PbBr$_4$ and (4-F/Cl/BrPMA)$_2$PbBr$_4$. This phenomenon could also be observed in the digital images of the samples (2-ClPMA)$_2$PbBr$_4$, (3-ClPMA)$_2$PbBr$_4$, (4-ClPMA)$_2$PbBr$_4$ under UV light (Fig. 2e), as well as the images of (F/BrPMA)$_2$PbBr$_4$ perovskites crystals (Supplementary Fig. 3).

Figure 3b shows the temperature-dependent PL spectra of (2-ClPMA)$_2$PbBr$_4$. The PL intensities of STEs and FEs both increase when temperature decreases from 298 to 77 K. The $I_{STEs}/I_{FEs}$ increases from about 2.9 at 298 K to 100 at 77 K (Fig. 3c), which is determined by a strong electron-phonon interaction induced transformation between FEs and STEs and PL quenching. At high temperature, it is more probable that FEs relax into STEs, however, the PL quenching is dominated due to the strong electron-phonon interaction. At low temperatures, the reverse applies. During the cooling process, a reduced electron-phonon interaction would narrow the full width at half maximum (FWHM) of both FEs and STEs emission (Fig. 3c). Figure 3d and Supplementary Fig. 4 illustrate the PL lifetime of (2-FPMA)$_2$PbBr$_4$, (2-ClPMA)$_2$PbBr$_4$, (3-ClPMA)$_2$PbBr$_4$ and (2-BrPMA)$_2$PbBr$_4$. The PL decay curves can be well fitted with a bi-exponential decay model. The PL lifetime is considered as a combination of a slow-decay component (radiative recombination) and a fast-decay component (nonradiative process) that give a long lifetime $\tau_1$ and a short lifetime $\tau_2$, respectively (Supplementary Table 6). According to previous report, the emission ratio $I_{STEs}/I_{FEs}$ at a given temperature is related to $\Delta G_{self-trap}$ (self-trapping depth $= E_{STEs} - E_{FEs}$) at emission peak and the radiative emission rates from the STEs and FEs states ($k_{r,STEs}$ and $k_{r,FEs}$)[19].

$$\ln\left(\frac{I_{STEs}}{I_{FEs}}\right) \propto \ln\left(\frac{k_{r,STEs}}{k_{r,FEs}}\right) - \frac{\Delta G_{self-trap}}{k_B T} \qquad (1)$$

Owing to the similar radiative PL lifetime and much lower energy of STEs state comparing to that of FEs state, the $I_{STEs}/I_{FEs}$ is dominated by the $\Delta G_{self-trap}$ (See Supplementary Table 6 for detailed discussion).

We further performed transient absorption (TA) studies to investigate the excited dynamics (Supplementary Fig. 5). In the

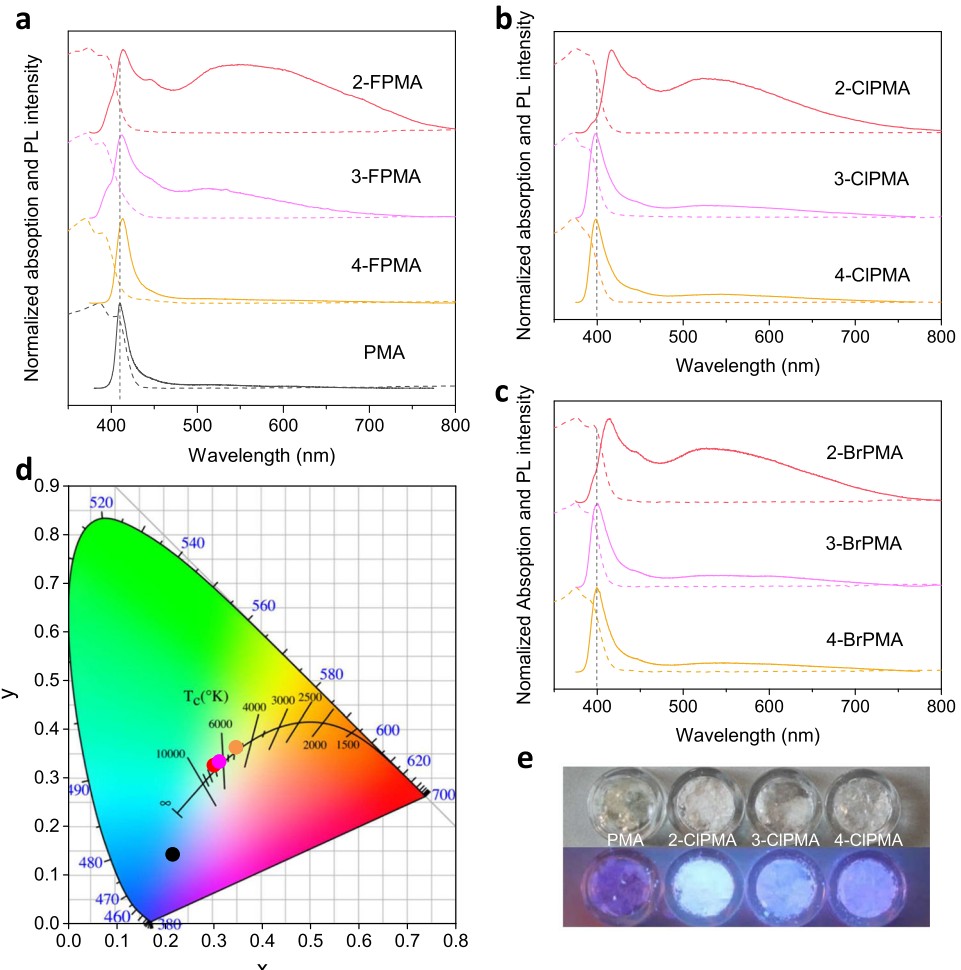

**Fig. 2 Optical properties of pristine and halogen-substituted perovskites. a–c** Absorption (dashed lines) and PL spectra (solid lines) of $(FPMA)_2PbBr_4$, $(ClPMA)_2PbBr_4$ and $(BrPMA)_2PbBr_4$. **d** CIE chromaticity coordinates of $(PMA)_2PbBr_4$ (black dot), $(2-FPMA)_2PbBr_4$ (yellow dot), $(2-ClPMA)_2PbBr_4$ (red dot) and $(2-BrPMA)_2PbBr_4$ (pink dot). **e** Digital images of $(PMA)_2PbBr_4$, $(2-ClPMA)_2PbBr_4$, $(3-ClPMA)_2PbBr_4$ and $(4-ClPMA)_2PbBr_4$ crystals under ambient light (upper) and UV light (lower).

perovskite $(2-FPMA)_2PbBr_4$, besides the almost opaque feature below 400 nm, we were able to observe two photo-induced absorption features located at about 405 nm and a broadband long-lived feature, which is likely related to the STE. The onsets of both photo-induced features are observed within the temporal resolution implying STE is probably formed simultaneously upon pulse excitation. The dynamics of the excited state exhibit a fast-decay component with lifetime of about 1 ps, which is possibly the feature of thermalization induced by exciton–phonon interaction. On the time scale of 10 ps, a slight delay rise component is captured, which may be attributed to the equilibrium buildup between FEs and STEs. Similar dynamics were also observed on $(2-Cl/BrPMA)_2PbBr_4$.

According to previous report, STEs emission correlates with increasing out-of-plane distortion of the Pb-Br-Pb angle in the inorganic sheets of the layered perovskites[19]. To investigated the structural origin of the STEs, a typical small group (-CH$_3$) ortho-substituted molecule 2-CH$_3$PMA was used to synthesize the perovskite $(2-CH_3PMA)_2PbBr_4$. The bond length of C-CH$_3$ is close to that of C-F and C-Cl. As expected, both powder XRD and single-crystal XRD measurements indicate $(2-CH_3PMA)_2PbBr_4$ has the nearly same crystal structure as $(2-FPMA)_2PbBr_4$ and $(2-ClPMA)_2PbBr_4$ (Supplementary Figs. 6, 7 and Supplementary Table 3). As shown in Supplementary Fig. 8, the out-of-plane Pb-

Br-Pb angles are all 180° for perovskites $(2-CH_3/F/ClPMA)_2PbBr_4$, meanwhile, their in-plane Pb-Br-Pb angles show only very small difference. However, we did not observe obvious broadband emission from $(2-CH_3PMA)_2PbBr_4$ (Supplementary Fig. 6). Inversely, the $(3-ClPMA)_2PbBr_4$ and $(4-ClPMA)_2PbBr_4$ without strong STE emission exhibit out-of-plane Pb-Br-Pb distortion. Thus, it can be concluded that emission features of these perovskites are ruled out of distortion origin of STEs as previous report and the crystal structure doesn't determine the formation of STEs in our case.

**The Coulomb force calculations.** As we know, polar crystal structures possess relatively strong electron-phonon interaction mainly owing to the Fröhlich contribution to the electron-phonon matrix elements[20]. The enhanced polarity of the components surrounding the excitons might thus promote STEs formation. Owing to the high electronegativity of halogen atoms, they are expected to withdraw electrons from the phenyl group and the branched chain -CH$_2$NH$_3^+$, inducing more positive charge accumulating on -CH$_2$NH$_3^+$. As shown in Fig. 4a, we evaluated the average Bader effective charges of -CH$_2$NH$_3^+$ in the unit cell by using first-principles density functional theory (DFT)-based calculations. The charges on halogen-substituted perovskites are clearly higher than those on pristine perovskite

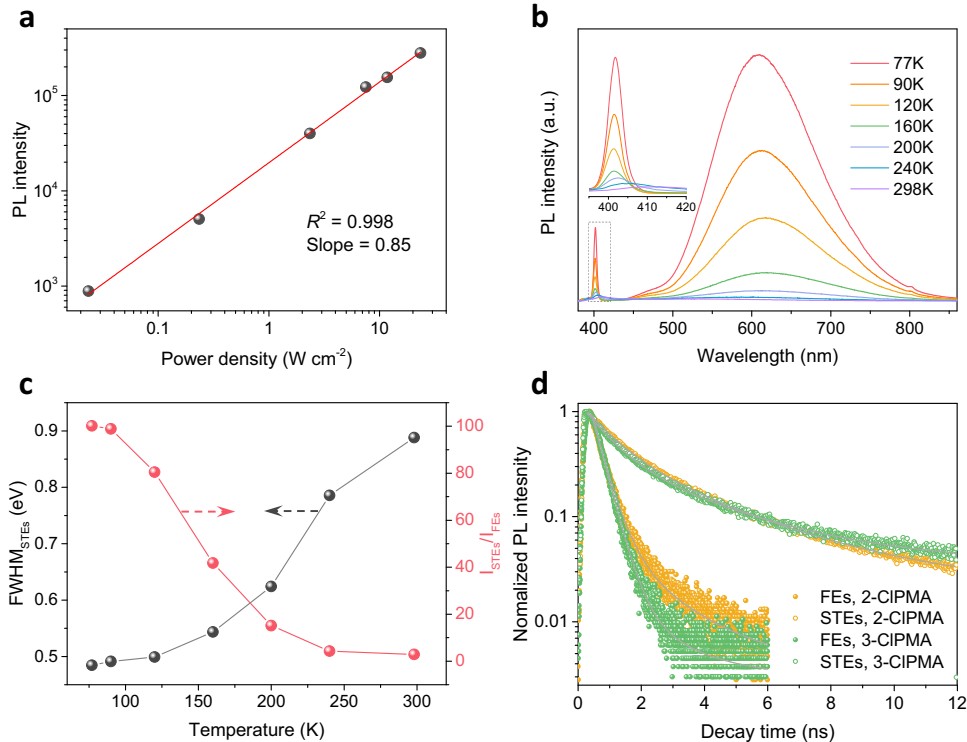

**Fig. 3 Photophysical properties of perovskites. a** Emission intensity versus excitation power for $(2\text{-ClPMA})_2\text{PbBr}_4$. The linear fit result has high $R^2$ value of 0.998. **b** Temperature-dependent PL spectra of $(2\text{-ClPMA})_2\text{PbBr}_4$. Inset is the zoom-in picture from 395 to 420 nm. **c** Temperature-dependent FWHM and ratio $I_{STEs}/I_{FEs}$ of $(2\text{-ClPMA})_2\text{PbBr}_4$ from 298 to 77 K. **d** PL decay profiles from FEs and STEs of $(2\text{-ClPMA})_2\text{PbBr}_4$ and $(3\text{-ClPMA})_2\text{PbBr}_4$.

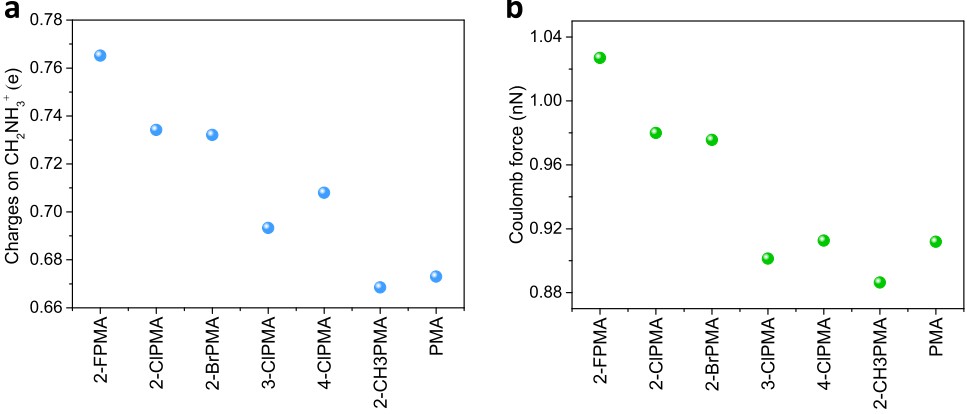

**Fig. 4 Charge characteristics of the perovskites. a** Calculated charge values of $-CH_2NH_3^+$ in pristine and substituted perovskites. **b** Calculated Coulomb force between $-CH_2NH_3^+$ and $PbBr_4^{2-}$ in pristine and substituted perovskites.

and $CH_3$-substituted perovskite. The orders of the charges values in halogen-substituted perovskites are ortho-substitution > para-substitution > meta-substitution and F-substitution > Cl-substitution ≈ Br-substitution. The average effective charge values of the $PbBr_4^{2-}$ anionic motifs were calculated in the same way. The Coulomb force was evaluated by assuming that the centers of the positive charges of $-CH_2NH_3^+$ cationic motifs and the negative charges of $PbBr_4^{2-}$ anionic motifs are located on the middle of C–N bond and Pb atoms, respectively. The Coulomb forces of the ionic bonds $(-CH_2NH_3^+)$-$(PbBr_4^{2-})$ were calculated to be 1.03, 0.98 and 0.98 nN for $(2\text{-FPMA})_2\text{PbBr}_4$, $(2\text{-ClPMA})_2\text{PbBr}_4$ and $(2\text{-BrPMA})_2\text{PbBr}_4$, which are higher than that of $(3\text{-ClPMA})_2\text{PbBr}_4$ $(4\text{-ClPMA})_2\text{PbBr}_4$, $(2\text{-CH}_3\text{PMA})_2\text{PbBr}_4$ and $(PMA)_2\text{PbBr}_4$ (Fig. 4b and Supplementary Table 7). Therefore, as expected the halogen substituents with high electronegativity indeed withdraw

electrons from the branched chain $(-R\text{-}NH_3^+)$ of the phenyl molecule. This results in a large positive charge accumulation on $-R\text{-}NH_3^+$, and thus a stronger attractive Coulomb force in ionic bond $(-R\text{-}NH_3^+)$-$(PbBr_4^{2-})$. Overall, the value of the Coulomb force is proportional to the $I_{STEs}/I_{FEs}$ ratio of perovskites. Thus, it can be convincingly speculated that the strengthened ionic bonds would trigger the STEs formation in the currently studied halide perovskites.

As the Coulomb force increasing, the electron-phonon interactions are expected to be enhanced, resulting in a recognized tendency of STEs formation. On the other hand, the strong electron-phonon interactions might lower the PLQY of the perovskites[16]. However, it may be understandable that the PLQY of the perovskites with STEs in this study have no simple proportional relationship to electron-phonon interactions and the

underlying Coulomb force, as other factors may affect the PLQY of the perovskites, such as trap species, trap density and potential dark excitons.

Moreover, we fabricated the phenylethylammonium lead bromide $((PEA)_2PbBr_4)$ and the corresponding halogen-substituted perovskites $(F/ClPEA)_2PbBr_4$. The broad emissions are also observed in $(2-F/ClPEA)_2PbBr_4$ (Supplementary Fig. 9), but their PL intensity ratios $I_{STEs}/I_{FEs}$ are much smaller than that of $(2-F/ClPMA)_2PbBr_4$. The PEA molecule has a longer branched chain ($-CH_2CH_2NH_3^+$) than PMA. Thus, the effect of electron-withdrawing from the branched chain to halogen substituents is weaker than that in PMA. As a result, the Coulomb force increase in halogen-substituted $(PEA)_2PbBr_4$ should be lower than that in halogen-substituted $(PMA)_2PbBr_4$.

## Stability of perovskite crystals

It is known that halide perovskites are sensitive to moisture, oxygen and heat. Thus, we took the stability measurement on the typical white perovskite (2-$ClPMA)_2PbBr_4$ by annealing the material on hotplate at 100 °C in $N_2$ glovebox or exposing the material in air with average temperature of 25 °C and humidity of 40%. As shown in Supplementary Fig. 10, the PLQY of the perovskite nearly remains the same even after continuous heating for 400 h. The material is also stable in air, with 85% of the initial PLQY remaining after 400 h. Powder XRD measurements also illustrated the structural stability of the perovskite after heating or exposing in air for 400 h (Supplementary Fig. 10).

## Discussion

In summary, we designed and synthesized novel halogen-substituted perovskite $(F/Cl/BrPMA)_2PbBr_4$ crystals by wet chemical methods. These perovskites display BWL emission with high PLQY, good CRI and excellent CIE coordinates close to standard white light (0.33, 0.33), indicating promising application for solid-state lighting. Combining first-principles calculations and experimental analysis, we discovered that the halogen substituents withdraw electrons from $-CH_2NH_3^+$, leading to an increase in the Coulomb force in $(-R-NH_3^+)-(PbBr_4^{2-})$ bonds and enhancement of broadband emission from STEs. The implication of our study is that the excitons emerging closely related to chemical bonds are of high importance for STEs formation, and new white perovskites may be designed and produced according to this strategy.

## Methods

**Materials and chemicals**. Lead bromide ($PbBr_2$) was purchased from Alfa Aesar. The pristine and halogen-substituted alkylamine, hydrobromic acid, N,N-dimethylformamide (DMF), dimethylsulfoxide (DMSO), chloroform were purchased from Aladdin. All chemicals were used as received.

**Synthesis of alkylamine bromide**. Pristine and halogen-substituted alkylamine bromide was prepared by adding alkylamine solution into hydrobromic acid (HBr, 48 wt% in water) with molar ratio 1.2: 1 at room temperature. The reaction mixture was stirred for 1 h. The solvent was removed via rotary evaporation at 50 °C. The raw product was washed with ethanol/diethyl ether for three times and dried at 60 °C in a vacuum oven for 12 h.

**Synthesis of perovskite crystals**. In all, 0.5 mmol $R-NH_3Br$ and 0.25 mmol $PbBr_2$ were dissolved in a mixed solvent of 0.25 mL DMF and 0.25 mL DMSO. In previous report, DMF/DMSO mixed solution is favorable for high-quality perovskite crystal growth[16]. Above perovskite precursor solution was put into a 5 mL vial without cap, which was put in a 30 mL vial with 10 mL anti-solvent chloroform. The larger vial was sealed and kept in an oven at 40 °C. Perovskite crystals grow gradually with anti-solvent diffusing into perovskite precursor solution. After few days, the final products were taken out and washed with anti-solvent for two times.

**Synthesis of perovskite powders by fast reprecipitation method**. In all, 0.5 mmol 2-ClPMABr and 0.25 mmol $PbBr_2$ were dissolved in a mixed solvent of 0.25 mL DMF and 0.25 mL DMSO. The precursor solution was drop into a vial containing 10 mL chloroform. The precipitation product was washed by chloroform for two times, and then dried at ambient atmosphere.

**Perovskite crystals characterizations**. Powder XRD measurements were performed using a Bruker D8 Advance diffractometer. Single-crystal X-ray diffraction experiments were carried out on a SuperNova diffractometer equipped with mirror Cu–Kα ($\lambda$ = 1.54184 Å) radiation and an Eos CCD detector under room temperature except for $(2-CH3PMA)_2PbBr_4$ (110 K). All the structures were solved by the direct method using the SHELXS program of the SHELXTL package and refined by the full-matrix least-squares method with SHELXL. The UV–Vis absorption was measured by using Shimadzu UV3600. Raman spectra were measured by using Renishaw inVia Raman microspectroscopy equipped with a 532 nm excitation laser. Steady-state PL and temperature-dependent PL spectra were measured by using Horiba HR Evolution spectrometer and a 365 nm LED lamp was used as the excitation source. A band-pass filter was applied to narrow the FWHM of the LED light to 10 nm. The power dependent PL intensity was recorded by using a femtosecond-pulse laser (NPI Laser, Rainbow 780, 80 MHz) with 390 nm output after passing through a double frequency crystal. Before measurements, we ground the crystals to avoid any angle dependence of incident/emissive light or heterogeneity between the connecting crystals. PLQY measurements were carried out by coupling an integrating sphere to fluorescent spectrometer (QEPro, Oceanoptics) with optical fibers. A 365 nm LED lamp was used as excitation source. Absolute irradiance was calibrated by a standard light source (HL-3plus, Oceanoptics). The accuracy of this system was verified by measuring Rhodamine 6G/ethanol solution (0.1 μmol/L), and the PLQY was measured to be 97 ± 2%, which is consistent with previous reports[21,22]. For lifetime measurements, single-photon counting was conducted with a PicoHarp 300 module (PicoQuant), excitation with the abovementioned 780 nm fs-pulsed laser. Two band-pass filters (405 and 618 nm) are used to separated decay dynamics for FEs and STEs, respectively. TA spectroscopy was conducted by employing a Ti:sapphire regenerative amplifier (Libra, Coherent Inc.) at 800 nm having a repetition rate of 1 kHz and pulse duration of 90 fs. An optical parametric amplifier (OperA Solo, Coherent Inc.) pumped by the regenerative amplifier was used to generate the pump beams (365 nm, about 7.5 μJ/cm$^2$). The probe beams were generated by focusing a small portion of the femtosecond laser beam onto a $CaF_2$ crystal. The TA signal was then analyzed by a silicon CCD (S11071, Hamamatsu) mounted on a monochromator (Acton 2358, Princeton Instrument) at 1 kHz enabled by a custom-built control board from Entwicklungsbuero Stresing. The pump beam modulated by an optical chopper at 500 Hz.

**First-principles calculations**. First-principles density functional theory (DFT)-based calculations were performed using the projector augmented wave (PAW)[23] potential method implemented in the Vienna Ab Initio Simulation Package code[24]. Generalized gradient approximation within the scheme of Perdew–Burke–Ernzerhof was chosen as the exchange-correlation functional[25]. The electron–ion interactions were described by using the PAW pseudopotentials with $5d^{10}6s^26p^2$(Pb), $4s^24p^5$(Br), $3s^23p^5$(Cl), $2s^22p^5$(F), $2s^22p^3$(N), $2s^22p^2$(C) and $1s^1$(H) treated explicitly as valence electrons. A Gamma-centered $k$-point mesh of 5×2×4 and an energy cutoff of 520 eV were adopted for electronic Brillouin zone integration. The perovskite structures (including lattice constants and atomic coordinates) were optimized through total energy minimization with residual forces on atoms converged below 0.01 eV Å$^{-1}$. To properly take into account the long-range van der Waals (vdWs) interaction that is non-negligible for hybrid perovskites involving organic molecules, the optB86b-vdW functional was adopted[26]. We used the Bader charge analysis method[27] to evaluate the average effective charges on the charged structural motifs (i.e., $-CH_2NH_3^-$ and $PbBr_4^{2-}$). The Bader charge analysis method divides the atoms by the zero-flux surfaces of the charge density and the charges on each atom are then determined by the summation of charges enclosed in the Bader volume[28]. The Coulomb attraction force of the ionic bond $(-CH_2NH_3^+)-(PbBr_4^{2-})$ was evaluated in term of the Coulomb law by assuming that the centers of the positive charges of $-CH_2NH_3^+$ cationic motifs and the negative charges of $PbBr_4^{2-}$ anionic motifs are located at the middle of C–N bond and Pb atoms, respectively. The first-principles calculations were performed in the automatic high-throughput mode by using the Jilin Artificial-intelligence aided Materials-design Integrated Package (JAMIP), which is an open-source artificial-intelligence-aided data-driven infrastructure designed purposely for computational materials informatics[29].

## Data availability

The data that support the findings of this study are available from the corresponding author upon reasonable request. The X-ray crystallographic coordinates for structures reported in this study have been deposited at the Cambridge Crystallographic Data Centre (CCDC), under deposition numbers 2076013-2076018. These data can be obtained free of charge from The Cambridge Crystallographic Data Centre via www.ccdc.cam.ac.uk/data_request/cif.

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

## Acknowledgements

J. Xing acknowledges financial support from National Natural Science Foundation of China (21905154) and the Taishan Scholars Program. W.X. acknowledges the Fundamental Research Funds for the Central Universities in China (020514380231), the National Natural Science Foundation of China (21873048), the Natural Science Foundation of Jiangsu Province (BK20180319), "Innovation & Entrepreneurship Talents Plan" of Jiangsu Province. L.Zhang acknowledges funding support from National Natural Science Foundation of China (Grant No. 92061113). Calculations were performed in part at the high-performance computing center of Jilin University.

## Author contributions

J. Xing conceived the idea and designed the experiments. M.Z. prepared the samples and performed the XRD, UV–Vis absorption measurements. L. Zhao and W.X. performed the optical measurements and analyzed the data. M.Z. analysied the results assisted by Q.Z., L.W. and W.X.. Z.Y. assisted to measured the PLQY. C.Z., S.Z., and H.L. conducted the TA measurements and analyzed the results. L. Zhang, J. Xie, Q.Z., X.W., N.Y., and P.K. did the first-principle calculations and analyzed the theoretical results. J. Xing and W.X. wrote the manuscript. All authors read and commented on the manuscript.

## Competing interests

The authors declare no competing interests.
