## [Peer Review File · Nature Communications]

REVIEWER COMMENTS

Reviewer #1 (Remarks to the Author):

In this work, the authors have reported an atom-substituting strategy to trigger STEs formation in layered perovskites through halogen substituting in intercalated organic molecules. The results are interesting but the scientific concerns and especially the novelty make it not quality for Nature Communication. I suggest the author to submit it elsewhere.

Here are some other detailed questions :

1. The authors attributed the formation of STE to the $\text{NH}_3^+-\text{PbBr}_3^-$ Coulomb attraction caused by the halogen-substitution, and gave the calculated average charge values of $-\text{NH}_3^+$ in different (CIPMA) $_2\text{PbBr}_4$ perovskites, however the detailed calculation method is not provided.
2. If the STE is triggered by the coulomb attraction as the authors explained, the (2-FPMA) $_2\text{PbBr}_4$ should show the strongest Coulomb force because of its higher electronegativity. The specific coulomb force of (2-FPMA) $_2\text{PbBr}_4$, (2-BrPMA) $_2\text{PbBr}_4$ should also be provided and compared with (2-CIPMA) $_2\text{PbBr}_4$. Besides, the corresponding ISTE/IFE of the (F/Cl/BrPMA) $_2\text{PbBr}_4$ should also be provided.
3. The excited state carrier dynamic of (2-F/Cl/BrPMA) $_2\text{PbBr}_4$ perovskites should also be provided.

Reviewer #2 (Remarks to the Author):

The paper sets out to investigate the origin for STEs formation in perovskites using atom substitution. The characterization work done is extensive and the results are well explained. There are several points that will need clarification and the link between the origin of STEs and changes in the coulomb force needs to be better shown.

1) Based on the charge quantity calculations, 2CIPMA is the highest (0.297), follow by 4CIPMA - 0.285 and 3CIPMA is 0.279. I don't see this translate to similar extend in the difference in coulomb force - the coulomb force is very similar for 3CIPMA (0.77nN) and 4CIPMA (0.78nN). What impact the coulomb force? Is there any distortion in angle between the octahedral? Are the degree of distortion change? Does distortion contribute to the observed coulomb force?

2) The PL for 2CH3PMA perovskite looks very similar the the 3Cl/4Cl system. What is the average charge for 2CH3PMA perovskite? How does this correlate to the coulomb force?

Reviewer #3 (Remarks to the Author):

In this manuscript, Zhang and coauthors report on halogen substituting hydrogen of C-H bond in 2D layered perovskites in order to induce the formation of self-trapped excitons. They found that the Coulomb force plays an important role in the formation of the self-trapped excitons. By adopting this atom-substituting strategy, they successfully achieved white perovskites with a high photoluminescence quantum yield of 32%. This study is interesting and important to 2D perovskite community. Therefore, I would like to recommend its publication after the authors successfully addressed my following concerns.

- 1 The authors need to provide more information on that hydrogen of C-H bond is indeed substituted by the halogen. There is no enough evidence to support this assumption. This is very important since all observations in the rest of manuscript are based on this assumption.
- 2 Even if the substitution doesnot take place, the induction of halogen atom would lead to the defect sites, which could also give rise to the extrinsic self-trapped excitons. How can the authors exclude this?
- 3 The authors are suggested to carry out single crystal XRD study, from which the lattice constant and distortion angle can be derived after halogen substitution.
- 4 In the powder dependent PL study, the range of the power used might be not large enough. If the amount of defects is large, linear powder dependent PL intensity can still be observed in if the power range is not large enough.

5 With the halogen substitution, the emission peak of free excitons seems also shift. Can the authors explain this? It is better to use vertical line to indicate the peak position in Figure 2 so that one can compare the emission peak among different samples.

6 For Figure 3a, the authors need to provide the slope of the line. If the slope deviates from one, the origin of the emission would be different.

7 There is no decisive conclusion that the excitons in 2D perovskites are Frenkel excitons. Rather, they have hybrid characteristics of both Frenkel and Wannier excitons. The authors are suggested to verify this.

8 The maximum PLQY the authors achieved is around 32%. Can the authors comment what are the limiting factors for further improving the PLQY?

9 Is it possible to achieve EL of self-trapped excitons in their samples?

Reviewer #1 (Remarks to the Author):

In this work, the authors have reported an atom-substituting strategy to trigger STEs formation in layered perovskites through halogen substituting in intercalated organic molecules. The results are interesting but the scientific concerns and especially the novelty make it not quality for Nature Communication. I suggest the author to submit it elsewhere. Here are some other detailed questions:

Response: We thank the reviewer for his/her valuable efforts in reviewing our manuscript and the positive assessment on our results. In response to the reviewer comments, we have performed new theoretical calculations and experimental characterizations to address the scientific concerns raised by the reviewer, and strengthened the novelty of our work in the revised manuscript.

Comment 1. The authors attributed the formation of STE to the $\text{NH}_3\text{-PbBr}_3$ Coulomb attraction caused by the halogen-substitution, and gave the calculated average charge values of $-\text{NH}_3^+$ in different $(\text{CIPMA})_2\text{PbBr}_4$ perovskites, however the detailed calculation method is not provided.

Response: Following the reviewer's suggestion, we have now provided more detailed calculation method information on the average charge values and the Coulomb attraction caused by the halogen-substitution in the Methods part of the revised Manuscript (Page 13).

Comment 2. If the STE is triggered by the coulomb attraction as the authors explained, the $(2\text{-FPMA})_2\text{PbBr}_4$ should show the strongest Coulomb force because of its higher electronegativity. The specific coulomb force of $(2\text{-FPMA})_2\text{PbBr}_4$, $(2\text{-BrPMA})_2\text{PbBr}_4$ should also be provided and compared with $(2\text{-CIPMA})_2\text{PbBr}_4$. Besides, the corresponding ISTE/IFE of the $(\text{F/Cl/BrPMA})_2\text{PbBr}_4$ should also be provided.

Response: We appreciate the reviewer for this constructive comment. In response, we have now performed additional new calculations of the Coulomb attraction force for different perovskites. In the previous Manuscript, we calculated the charges and Coulomb force on the substituted perovskites by using the crystal structures from theoretical simulation. In the

revised version, we took single-crystal XRD measurements on these perovskites and obtained explicit lattice parameters and internal atomic coordinates of crystal structures. We then recalculated the charges and Coulomb force based on the experimental structures. After carefully analyzing the average charge values of the charged structural motifs, we realized that the $-\text{CH}_2\text{NH}_3^+$ group is more reasonably regarded as the effective charged motif to evaluate the Coulomb attraction force. This is because that the similar electronegativities of carbon and nitrogen make it hard to accurately determine the internal charge attribution within the $-\text{CH}_2\text{NH}_3^+$ group. We thus evaluated the average charge values of the $-\text{CH}_2\text{NH}_3^+$ cationic motifs upon halogen substitution by employing the state-of-the-art Bader charge population method, which have been proven as an effective approach for calculating charge population in complex material systems [Nature 2016, 537, 382; Nat. Chem. 2015, 7, 250; J. Am. Chem. Soc. 2016, 138, 6028]. As shown in the newly added Fig. 4 in the revised manuscript (shown also as below Fig. R1-1), we obtained the average charge values of $-\text{CH}_2\text{NH}_3^+$ in the unit cell. The charges on halogen-substituted perovskites are higher than those on pristine perovskite and CH_3 -substituted perovskite. The orders of the charges values in halogen-substituted perovskites are ortho-substitution > para-substitution > meta-substitution and F-substitution > Cl-substitution \approx Br-substitution. The average charge values of the PbBr_4^{2-} anionic motifs were calculated in the same way. The Coulomb attraction force were evaluated by assuming that the centers of the positive charges of $-\text{CH}_2\text{NH}_3^+$ cationic motifs and the negative charges of PbBr_4^{2-} anionic motifs are located on the middle of C-N bond and Pb atoms, respectively. The Coulomb forces of the ionic bonds ($-\text{CH}_2\text{NH}_3^+$)-(PbBr_4^{2-}) were calculated to be 1.03, 0.98 and 0.98 nN for $(2\text{-FPMA})_2\text{PbBr}_4$, $(2\text{-CIPMA})_2\text{PbBr}_4$ and $(2\text{-BrPMA})_2\text{PbBr}_4$, which are higher than that of $(3\text{-CIPMA})_2\text{PbBr}_4$, $(4\text{-CIPMA})_2\text{PbBr}_4$, $(2\text{-CH}_3\text{PMA})_2\text{PbBr}_4$ and $(\text{PMA})_2\text{PbBr}_4$. Therefore, as expected the halogen-substituents with high electronegativity indeed withdraw electrons from the branched chain ($-\text{R-NH}_3^+$) of the phenyl molecule. This results in a large positive charge accumulation on $-\text{R-NH}_3^+$, and thus a stronger attractive Coulomb force in ionic bond ($-\text{R-NH}_3^+$)-(PbBr_4^{2-}). Overall, the value of the Coulomb force is proportional to the $I_{\text{STES}}/I_{\text{FES}}$ ratio of perovskites (the $I_{\text{STES}}/I_{\text{FES}}$ ratios of perovskites were now provided in the revised Manuscript, page 7). Therefore, it can be convincingly speculated that the strengthened ionic

bonds would trigger the STEs formation in perovskites.

We have revised the manuscript based on the above newly added results and discussion (Page 10).

Fig. R1-1. (a) Calculated charge values of $-\text{CH}_2\text{NH}_3^+$ in pristine and substituted perovskites. (b) Calculated Coulomb force between $-\text{CH}_2\text{NH}_3^+$ and PbBr_4^{2-} in pristine and substituted perovskites.

Comment 3. The excited state carrier dynamic of $(2\text{-F/Cl/BrPMA})_2\text{PbBr}_4$ perovskites should also be provided.

Response: We have provided the excited state carrier dynamic of $(2\text{-F/Cl/BrPMA})_2\text{PbBr}_4$ perovskites and summarized the decay parameters in the revised Manuscript (Fig. 3d) and Supplementary Information (Supplementary Fig. 4, Supplementary Table 6), which are also shown as below Figure 1-2 and Table 1-1. The PL decay curves can be well fitted with a bi-exponential decay model. The PL lifetime is considered as a combination of a slow-decay component and a fast-decay component that give a long lifetime τ_1 and a short lifetime τ_2 , respectively. Corresponding discussion texts have also been added in the revised Manuscript (Page 8).

Fig. R1-2. PL decay profiles from FEs and STEs of $(2\text{-CIPMA})_2\text{PbBr}_4$, $(3\text{-CIPMA})_2\text{PbBr}_4$ (left) and $(2\text{-FPMA})_2\text{PbBr}_4$, $(3\text{-BrPMA})_2\text{PbBr}_4$.

Table R1-1. The PL decay parameters of $(2\text{-FPMA})_2\text{PbBr}_4$, $(2\text{-CIPMA})_2\text{PbBr}_4$, $(3\text{-CIPMA})_2\text{PbBr}_4$ and $(2\text{-BrPMA})_2\text{PbBr}_4$.

	$\tau_{1,\text{STEs}}$ (ns)	$\tau_{2,\text{STEs}}$ (ns)	$\tau_{1,\text{FEs}}$ (ns)	$\tau_{2,\text{FEs}}$ (ns)
$(2\text{-FPMA})_2\text{PbBr}_4$	2.72	0.73	1.30	0.36
$(2\text{-CIPMA})_2\text{PbBr}_4$	3.31	0.93	1.58	0.38
$(3\text{-CIPMA})_2\text{PbBr}_4$	3.23	0.84	1.50	0.35
$(2\text{-BrPMA})_2\text{PbBr}_4$	2.08	0.54	1.60	0.41

In response to the reviewer’s comment on the novelty of our work, we want to emphasize that in this work we discovered a never-reported-before novel atom-substituting strategy to surprisingly trigger formation of self-trapped excitons (STE) in two-dimensional layered perovskites. The underlying physical mechanism was unraveled by the joint experiment-theory study, which was attributed to the halogen-substituting-hydrogen induced enhancement of attractive Coulomb force between cationic organic motifs and anionic perovskite framework. The enhanced ionic bonding affects significantly electron-phonon interaction profile and increases possibility of exciton self-trapping. For the first time our work points out a novel way of modulating STE by deliberately engineering the strength of ionic bonding, and provide new insight into understanding the STE formation in low-dimensional perovskites. This may be exploited to further improve the light emission

performance of STE and rationally design new STE materials based on the low-dimensional perovskites.

We appreciate the Reviewer for offering us such an opportunity to further improve the manuscript quality.

Reviewer #2 (Remarks to the Author):

The paper sets out to investigate the origin for STEs formation in perovskites using atom substitution. The characterization work done is extensive and the results are well explained. There are several points that will need clarification and the link between the origin of STEs and changes in the coulomb force needs to be better shown.

Response: We thank the reviewer very much for the positive comments. We have responded to detailed comments as follows and revised the Manuscript accordingly.

Comment 1. Based on the charge quantity calculations, 2CIPMA is the highest (0.297), follow by 4CIPMA - 0.285 and 3CIPMA is 0.279. I don't see this translate to similar extend in the difference in coulomb force - the coulomb force is very similar for 3CIPMA (0.77nN) and 4CIPMA (0.78nN). What impact the coulomb force? Is there any distortion in angle between the octahedral? Are the degree of distortion change? Does distortion contribute to the observed coulomb force?

Response: According to the Coulomb law $F = k \frac{q_1 \times q_2}{d^2}$, the Coulomb force was determined by both charge quantity and distance between positive and negative charges. Although the average charge quantity of -NH_3^+ in 4-CIPMA is larger than that in 3-CIPMA, the average distance of $\text{-NH}_3^+ \text{-PbBr}_4^{2-}$ ionic bond in 4-CIPMA is longer than that in 3-CIPMA. Finally, the resulted Coulomb forces of 4-CIPMA and 3-CIPMA are close.

According to previous report (Chem. Sci. 2017, 8, 4497), STE emission correlates with increasing out-of-plane distortion of the Pb-Br-Pb angle in the inorganic sheets of the layered perovskites. In the revised version, we took single-crystal XRD measurements on these perovskites and obtained explicit lattice parameters and internal atomic coordinates of crystal structures. This allows us to accurately evaluate the distortion in angle between the octahedra. As shown in Fig. R2-1 (shown also as Supplementary Fig. 7 in Supplementary Information), the out-of-plane Pb-Br-Pb angles are all 180° for perovskites $(2\text{-CH}_3/\text{F}/\text{Cl}/\text{BrPMA})_2\text{PbBr}_4$, meanwhile, their in-plane Pb-Br-Pb angles show only very small difference. However, we did not observe obvious broadband emission from $(2\text{-CH}_3\text{PMA})_2\text{PbBr}_4$. Inversely, the $(3\text{-CIPMA})_2\text{PbBr}_4$ and $(4\text{-CIPMA})_2\text{PbBr}_4$ without strong STE emission exhibit out-of-plane

Pb–Br–Pb distortion. These results indicate that that these perovskites are ruled out of distortion origin of STEs as previous report.

For Coulomb force calculations, the distortion in angle between the octahedra in perovskites would influence the distance between $-\text{NH}_3^+-\text{PbBr}_4^{2-}$ slightly (see Supplementary Table 7 in Supplementary Information), but would not have substantial impact on the charges on $-\text{NH}_3^+$. Overall, the distortion is not a decisive factor for the Coulomb force and the formation of STEs.

Fig. R2-1. The structure of out-of-plane and in-plane distortion of perovskites.

Comment 2. The PL for 2CH3PMA perovskite looks very similar the 3Cl/4Cl system. What is the average charge for 2CH3PMA perovskite? How does this correlate to the coulomb force?

Response: In the previous Manuscript, we calculated the charges and Coulomb force on the substituted perovskites by using the crystal structures from theoretical simulation. In the

revised version, we took single-crystal XRD measurements on these perovskites and obtained explicit lattice parameters and internal atomic coordinates of crystal structures. We then recalculated the charges and Coulomb force based on the experimental structures. After carefully analyzing the average charge values of the charged structural motifs, we realized that the $-\text{CH}_2\text{NH}_3^+$ group is more reasonably regarded as the effective charged motif to evaluate the Coulomb attraction force. This is because that the similar electronegativities of carbon and nitrogen make it hard to accurately determine the internal charge attribution within the $-\text{CH}_2\text{NH}_3^+$ group. We thus evaluated the average charge values of the $-\text{CH}_2\text{NH}_3^+$ cationic motifs upon halogen substitution by employing the state-of-the-art Bader charge population method, which have been proven as an effective approach for calculating charge population in complex material systems [Nature 2016, 537, 382; Nat. Chem. 2015, 7, 250; J. Am. Chem. Soc. 2016, 138, 6028]. As shown in the newly added Fig. 4 in the revised manuscript (shown also as below Fig. R2-2), we obtained the average charge values of $-\text{CH}_2\text{NH}_3^+$ in the unit cell. The charges on halogen-substituted perovskites are higher than those on pristine perovskite and CH_3 -substituted perovskite. The orders of the charges values in halogen-substituted perovskites are ortho-substitution > para-substitution > meta-substitution and F-substitution > Cl-substitution \approx Br-substitution. The average charge values of the PbBr_4^{2-} anionic motifs were calculated in the same way. The Coulomb attraction force were evaluated by assuming that the centers of the positive charges of $-\text{CH}_2\text{NH}_3^+$ cationic motifs and the negative charges of PbBr_4^{2-} anionic motifs are located on the middle of C-N bond and Pb atoms, respectively. The Coulomb forces of the ionic bonds ($-\text{CH}_2\text{NH}_3^+$)-(PbBr_4^{2-}) were calculated to be 1.03, 0.98 and 0.98 nN for $(2\text{-FPMA})_2\text{PbBr}_4$, $(2\text{-CIPMA})_2\text{PbBr}_4$ and $(2\text{-BrPMA})_2\text{PbBr}_4$, which are higher than that of $(3\text{-CIPMA})_2\text{PbBr}_4$, $(4\text{-CIPMA})_2\text{PbBr}_4$, $(2\text{-CH}_3\text{PMA})_2\text{PbBr}_4$ and $(\text{PMA})_2\text{PbBr}_4$. Therefore, as expected the halogen-substituents with high electronegativity indeed withdraw electrons from the branched chain ($-\text{R-NH}_3^+$) of the phenyl molecule. This results in a large positive charge accumulation on $-\text{R-NH}_3^+$, and thus a stronger attractive Coulomb force in ionic bond ($-\text{R-NH}_3^+$)-(PbBr_4^{2-}). Overall, the value of the Coulomb force is proportional to the $I_{\text{STEs}}/I_{\text{FES}}$ ratio of perovskites. Therefore, it can be convincingly speculated that the strengthened ionic bonds would trigger the STEs formation in perovskites.

Fig. R2-2. (a) Calculated charge values of $-\text{CH}_2\text{NH}_3^+$ in pristine and substituted perovskites. (b) Calculated Coulomb force between $-\text{CH}_2\text{NH}_3^+$ and PbBr_4^{2-} in pristine and substituted perovskites.

Reviewer #3 (Remarks to the Author):

In this manuscript, Zhang and coauthors report on halogen substituting hydrogen of C-H bond in 2D layered perovskites in order to induce the formation of self-trapped excitons. They found that the Coulomb force plays an important role in the formation of the self-trapped excitons. By adopting this atom-substituting strategy, they successfully achieved white perovskites with a high photoluminescence quantum yield of 32%. This study is interesting and important to 2D perovskite community. Therefore, I would like to recommend its publication after the authors successfully addressed my following concerns.

Response: We thank the reviewer for the positive comments. In response to the reviewer's constructive comments, we have performed new experiments and made extensive revision to strengthen the manuscript.

Comment 1. The authors need to provide more information on that hydrogen of C-H bond is indeed substituted by the halogen. There is not enough evidence to support this assumption. This is very important since all observations in the rest of manuscript are based on this assumption.

Response: We thank the reviewer for making this good comment. We fabricated the halogen-substituted perovskites by two-steps. Firstly, we synthesized the halogen-substituted alkylammonium bromides via reaction of the halogen-substituted alkylamines and hydrobromic acid at room temperature. The halogen-substituted alkylamines are commercial chemicals with purity of 97-99%, they were purchased from Aladdin. The C-F and C-Cl bonds are much stronger than C-Br bond, the Br from HBr will not replace the F or Cl on F/Cl-substituted alkylamines under mild condition. Secondly, we synthesized the perovskite crystals by facile anti-solvent diffusion method at 40 °C, which couldn't break the strong covalence bonds C-halogen.

To further verify this point, we took Raman measurements on the perovskites and their corresponding alkylamines precursors. The obtained Raman features of alkylamines precursors are all same as the Raman data provided by Sigma-Aldrich (<https://www.sigmaaldrich.com/>). As shown Fig. R3-1 (shown also as Supplementary Fig. 1

in the Supplementary Information), Raman spectra of all the perovskites can match well with their corresponding precursors. For the halogen-substituted perovskites, the typical Raman band at about 680 cm^{-1} is attributed to typical halogen-substituent-sensitive vibration (J. Mol. Spectrosc. 1971, 39, 73-78), which is not observed on pristine perovskite. Meanwhile, this Raman band redshifts as the electronegativity of halogen substituent decrease that is consistent with previous report (J. Mol. Spectrosc. 1971, 39, 73-78). Therefore, halogen-substituted perovskites were successfully synthesized and the halogen substituents can stably stay in the perovskite products.

We provided this data in the revised Manuscript.

Fig. R3-1. Raman spectra of perovskite $(\text{PMA})_2\text{PbBr}_4$, $(2\text{-FPMA})_2\text{PbBr}_4$, $(2\text{-CIPMA})_2\text{PbBr}_4$, $(2\text{-BrPMA})_2\text{PbBr}_4$, $(3\text{-CIPMA})_2\text{PbBr}_4$, $(4\text{-CIPMA})_2\text{PbBr}_4$ (solid lines) and their corresponding alkylamines precursors purchased from Aladdin (dashed lines).

Comment 2. Even if the substitution does not take place, the induction of halogen atom would lead to the defect sites, which could also give rise to the extrinsic self-trapped excitons. How can the authors exclude this?

Response: As mentioned above, the halogen substituents stably stay on the molecular of the perovskites, which would not induce defect sites as dopant.

Comment 3. The authors are suggested to carry out single crystal XRD study, from which the

lattice constant and distortion angle can be derived after halogen substitution.

Response: As reviewer's suggestion, we carried out the single-crystal XRD measurement and provided the lattice constant and distortion angle in Supplementary Information (Supplementary Table 2, 3 and Supplementary Fig. 7). According to previous report (Chem. Sci. 2017, 8, 4497), the out-of-plane distortion in perovskite might induce the STE formation. However, in the perovskites $(\text{PMA})_2\text{PbBr}_4$, $(2\text{-FPMA})_2\text{PbBr}_4$, $(2\text{-CIPMA})_2\text{PbBr}_4$ and $(2\text{-BrPMA})_2\text{PbBr}_4$, there are nearly no distortion in out-of-plane orientation (Fig. R3-2). Furthermore, the structure of $(2\text{-FPMA})_2\text{PbBr}_4$ with obvious STE emission is very close to that of $(2\text{-CH}_3\text{PMA})_2\text{PbBr}_4$ without obvious STE emission. Thus, we can exclude that structural distortion origin of the STE formation in the halogen-substituted perovskites. We supplemented these data in the revised Manuscript (Page 9).

Fig. R3-2. The structure of out-of-plane (upper) and in-plane (lower) distortion of perovskites.

Comment 4. In the powder dependent PL study, the range of the power used might be not large enough. If the amount of defects is large, linear powder dependent PL intensity can still be observed in if the power range is not large enough.

Response: Emission from permanent material defects typically show a sublinear dependence on excitation power density with a saturation of limited defect sites under high excitation intensity. For linear power dependent emission from permanent defects: excitation rate \ll relaxation rate (Phys. Rev. B 2001, 64, 115205; J. Am. Chem. Soc. 2014, 136, 13154).

σ = absorption cross section, A = absorbance, ϵ_M = molar extinction coefficient, L = film thickness, C = concentration, N_A = Avogadro's number, q_p = laser photon flux, I = laser intensity, h = Planck's constant, c = velocity of light, λ = laser wavelength (390 nm). To obtain the absorption parameters of sample, we fabricated a (2-CIPMA)₂PbBr₄ perovskite film by spincoating method. Its thickness is about 100 nm and absorbance at 390 nm is about 0.8.

Excitation rate = σq_p

$A = \epsilon_M \times C \times L$ ($C = 0.0030 \text{ mol/cm}^3$ obtained from formula units per unit cell volume)

$\epsilon_M = A / (C \times L) = 2.67 \times 10^7 \text{ cm}^2/\text{mol}$

$\sigma = \ln(10) \times (\epsilon_M / N_A) = 1.02 \times 10^{-16} \text{ cm}^2/\text{Pb}^{2+}$

$I = 23.5 \text{ W/cm}^2$ ($2.45 \times 10^6 \text{ W/cm}^2$ for one pulse)

$q_p = I\lambda/hc = 4.81 \times 10^{24} \text{ s}^{-1} \text{ cm}^{-2}$

$\sigma q_p = (1.02 \times 10^{-16} \text{ cm}^2/\text{Pb}^{2+}) \times (4.81 \times 10^{24} \text{ s}^{-1} \text{ cm}^{-2}) \approx 5 \times 10^8 \text{ (s}^{-1}/\text{Pb}^{2+})$

Relaxation rate = $N/(\text{PL lifetime})$

N = number of emissive defects per Pb^{2+} . PL lifetime at 298 K $\sim 10^{-9}$ s

If emission arises from permanent material defects, PL unsaturation at laser power density of 23.5 W/cm^2 occurs only when Excitation rate \ll relaxation rate.

$5 \times 10^8 \text{ (s}^{-1}/\text{Pb}^{2+}) \ll N/(10^{-9} \text{ s})$

So, $N \gg 0.5 \text{ defects}/\text{Pb}^{2+}$ or $N' \gg 10^{21} \text{ defects}/\text{cm}^3$.

According to previous reports, the defects density of perovskite single-crystal is usually 10^9 - $10^{10}/\text{cm}^3$ (Science 2015, 347, 967; Science 2015, 347, 519). The defects density of $10^{21}/\text{cm}^3$ ($0.5 \text{ defects}/\text{Pb}^{2+}$) could be unlikely to occur in our materials. Thus, the power density of 23.5 W/cm^2 should be high enough to exclude that the broadband emission originate from permanent defects.

Moreover, we proved the STEs emission of the perovskites by comparing the PL spectra of (2-CIPMA)₂PbBr₄ crystals synthesized by fast reprecipitation and slow anti-solvent diffusion methods. The polycrystals synthesized by fast reprecipitation method (crystallization period is less than one second) should have more defects than that of the single-crystal synthesized by slow anti-solvent diffusion method (crystallization period is

several days). However, their emitting spectra are nearly the same. Therefore, we can conclude that the broadband emission originates from STEs rather than permanent defects.

We have added above analysis in the Supplementary Information.

Comment 5. With the halogen substitution, the emission peak of free excitons seems also shift. Can the authors explain this? It is better to use vertical line to indicate the peak position in Figure 2 so that one can compare the emission peak among different samples.

Response: The emission peak of the free excitons would be influenced by multiple factors. Firstly, the halogen ortho/meta/para-substituted perovskites have slightly different bandgap with each other. Secondly, the range of PL Stokes-shift of these perovskites might also be different. Finally, strong reabsorption of high energy region of the free excitons emission would occur in the bulk crystals, which may lead to a dominative PL peak redshifting. As reviewer's suggestion, we use vertical line to indicate the peak position as shown in Fig. R3-3 and explained in the revised Manuscript (Page 7).

Fig. R3-3. Absorption (dashed lines) and PL spectra (solid lines) of $(\text{FPMA})_2\text{PbBr}_4$, $(\text{CIPMA})_2\text{PbBr}_4$ and $(\text{BrPMA})_2\text{PbBr}_4$.

Comment 6. For Figure 3a, the authors need to provide the slope of the line. If the slope deviates from one, the origin of the emission would be different.

Response: As the reviewer's suggestion, we provided the slope in Fig. 3a. The slope is 0.85, which is close to 1.

Comment 7. There is no decisive conclusion that the excitons in 2D perovskites are Frenkel excitons. Rather, they have hybrid characteristics of both Frenkel and Wannier excitons. The authors are suggested to verify this.

Response: Photoinduced excitons in low-dimensional perovskites have large binding energies of several hundreds of meV, which are very like Frenkel excitons. To avoid the dispute of Frenkel and Wannier excitons in 2D perovskite, we don't classify the excitons in 2D perovskite, and revised the statement as follows. This revision does not affect the discussion and conclusions of this work.

“Photoinduced excitons in low-dimensional perovskites have large binding energies of several hundreds of meV, in which an electron-hole pair sharing common bonds.”

Comment 8. The maximum PLQY the authors achieved is around 32%. Can the authors comment what are the limiting factors for further improving the PLQY?

Response: Multiple factors may influence PLQY of the 2D perovskites. According to previous report (Nat. Mater. 2018, 17, 550), the PLQY of 2D perovskite is not only determined by the quantity of defects, but also affected by electron-phonon interaction. (1) The quantity of defects in perovskite crystal could be reduced by optimizing the crystal growth condition. In this work, we have optimized the crystal growth condition to improve the PLQYs of the perovskites, though it is probably not the optimal condition for synthesizing the best perovskites. (2) Electron-phonon interactions are influenced by the structure of organic component of 2D perovskites. Here, strong electron-phonon coupling is like a two-edged sword for white perovskites, which is necessary for STE formation, but may result in fast non-radiative decay, lowering the PLQY.

Comment 9. Is it possible to achieve EL of self-trapped excitons in their samples?

Response: We have tried hard to fabricate the (2-F/Cl/BrPMA)PbBr₄ based LEDs with architecture of ITO/PEDOT:PSS/perovskite film/TPBi/LiF/Al. However, the LEDs don't work. It might be because of (1) the charge mobility of layered perovskite is very low, electrons and holes are difficult to inject into the perovskite layer of LEDs; (2) the very low

PLQYs of the spincoated perovskite films induce severe nonradiative recombination losses.
In the future, we will still work on white-light perovskite based LEDs.

REVIEWER COMMENTS

Reviewer #1 (Remarks to the Author):

Below are some points which I think are worth to address. With this future improvements, this manuscript may be suitable for publication.

- 1) Coulomb force will influence the formation time of self-trapping? I suggest the author to provide transit absorption results.
- 2) However Coulomb force influence PLQY? I suggest the author to add more discussions.
- 3) How about the stability of these materials?

Reviewer #2 (Remarks to the Author):

I think the replies provide the authors addressed most of the concerns. I think the novelty of the work which is questioned by the first reviewer, is still not addressed.

Reviewer #3 (Remarks to the Author):

The authors have successfully addressed my questions and revised the manuscript accordingly. Therefore, I would like to recommend its publication.

Reviewer #1 (Remarks to the Author):

Below are some points which I think are worth to address. With this future improvements, this manuscript may be suitable for publication.

Response: We appreciate the reviewer for raising new constructive comments to help us further improve the quality of the manuscript. In response, we have performed additional experimental characterizations and revised the manuscript by adding the newly obtained data.

Comment 1. Coulomb force will influence the formation time of self-trapping? I suggest the author to provide transit absorption results.

Response: We thank the reviewer for this good comment. We performed transient absorption (TA) studies to investigate the excited dynamics (Fig. R1, also shown as Supplementary Fig. 5 in Supplementary Information). In the perovskite $(2\text{-FPMA})_2\text{PbBr}_4$, besides the almost opaque feature below 400 nm, we were able to observe two photo-induced absorption features located at about 405 nm and a broadband long-lived feature which is likely related to the STE. The onsets of both photo-induced features are observed within the temporal resolution implying STE is probably formed simultaneously upon pulse excitation. The dynamics of the excited state exhibit a fast decay component with lifetime of ~ 1 ps which is possibly the feature of thermalization induced by exciton-phonon interaction. On the time scale of 10 ps, a slight delay rise component is captured, which may be attributed to the equilibrium buildup between FEs and STEs. Similar dynamics were observed on perovskites $(2\text{-Cl/BrPMA})_2\text{PbBr}_4$. Overall, STEs form very fast in these perovskites and the Coulomb force has negligible influence on the formation time. Corresponding discussion has also been added in the revised Manuscript (Page 9).

Fig. R1. (a) 2D color plots of TA spectra of (2-FPMA)₂PbBr₄. (b) TA time delay of perovskites (2-F/Cl/BrPMA)₂PbBr₄ probed at 405 and 575 nm.

Comment 2. However Coulomb force influence PLQY? I suggest the author to add more discussions.

Response: Following the reviewer’s suggestion, we have now added the following discussion into Page 11. “As the Coulomb force increasing, the electron-phonon interactions are expected to be enhanced, resulting in recognized tendency of STEs formation. On the other hand, the strong electron-phonon interactions might lower the PLQY of the perovskites [Nat. Mater. 2018, 17, 550]. However, it may be understandable that the PLQY of the perovskites with STEs in this study have no simple proportional relationship to electron-phonon interactions and the underlying Coulomb force (see Fig. R2), as other factors may affect the PLQY of the perovskites, such as trap species, trap density and potential dark excitons.”

Fig. R2. PLQYs and Coulomb forces of perovskites $(PMA)_2PbBr_4$ and $(2-F/Cl/BrPMA)_2PbBr_4$.

Comment 3. How about the stability of these materials?

Response: We thank the reviewer for making the comment. It is known that halide perovskites are sensitive to moisture, oxygen and heat. During this revision, we took the stability measurements on the typical white perovskite $(2-CIPMA)_2PbBr_4$ by annealing the material on hotplate at $100\text{ }^\circ\text{C}$ in N_2 glovebox and exposing the material in air with average temperature $25\text{ }^\circ\text{C}$ and humidity 40%. As shown in Fig. R3 (shown as Supplementary Fig. 10 in Supplementary Information), the PLQY of the perovskite nearly remains the same even after continuous heating for 400 hours. The material is also stable in air, with 85% of the initial PLQY remaining after 400 hours. Powder XRD measurements also illustrated the structural stability of the perovskite after heating or exposing in air for 400 hours. We added these data in the revised manuscript (Page 12).

Fig. R3. (a) Stability of the perovskite $(2-CIPMA)_2PbBr_4$ against continuous heating at $100\text{ }^\circ\text{C}$

on hotplate in N₂ glovebox and exposing in air (average temperature and humidity, 25 °C and 40%). (b) XRD patterns of the perovskite (2-CIPMA)₂PbBr₄ after stability measurements.

Reviewer #2 (Remarks to the Author):

I think the replies provide the authors addressed most of the concerns. I think the novelty of the work which is questioned by the first reviewer, is still not addressed.

Response: We thank the reviewer again for the valuable efforts and for raising the constructive comments to help us improve the quality of the manuscript. Regarding the novelty of the work which is questioned by the first reviewer, we want to emphasize that because of poor understanding of the STEs formation process and its underlying physical mechanism in perovskites, it is still a big challenge to design new perovskites with efficient STEs for white light emission applications.

In this work, we developed a never-reported-before novel atom-substituting strategy to surprisingly trigger self-trapped excitons (STE) formation in low-dimensional perovskites. The underlying physical mechanism was unraveled by the joint experiment-theory study, which was attributed to the halogen-substituents induced enhancement of attractive Coulomb force between cationic organic motifs and anionic inorganic perovskite framework. The enhanced ionic bonding affects significantly electron-phonon interaction and increases possibility of exciton self-trapping. For the first time, our work points out a novel way of modulating STE by deliberately engineering the strength of ionic bonding, and provide new insight into understanding the STE formation in low-dimensional perovskites.

In the newly revised manuscript, transient absorption profiles provide new insight into the dynamics of STEs. We observed the decay process of STEs in the perovskite.

Overall, this work not only discovers a series of new perovskites emitting broadband white-light with high quantum efficiency and good stability, but unravels the effect of the chemical bonds strength on the STEs formation, and thus offers a useful perspective to design new white perovskite materials.

We have now highlighted the above points in the Abstract and Introduction sections of the revised manuscript. We appreciate the Reviewer for offering us such an opportunity to improve the manuscript.

Reviewer #3 (Remarks to the Author):

The authors have successfully addressed my questions and revised the manuscript accordingly.

Therefore, I would like to recommend its publication.

Response: We thank the reviewer for recommending the manuscript's publication.

REVIEWERS' COMMENTS

Reviewer #1 (Remarks to the Author):

All my previous concerns are well addressed now; it is ready for publication.

Reviewer #3 (Remarks to the Author):

The authors have successfully addressed my questions and revised the manuscript accordingly.

Therefore, I would like to recommend its publication.

Response: We thank the reviewer for recommending the manuscript's publication.